# Construction of Chitosan/Alginate Nano-Drug Delivery System for Improving Dextran Sodium Sulfate-Induced Colitis in Mice

**DOI:** 10.3390/nano11081884

**Published:** 2021-07-22

**Authors:** Mengfei Jin, Shangyong Li, Yanhong Wu, Dandan Li, Yantao Han

**Affiliations:** School of Basic Medicine, Qingdao University, Ningxia Road 308, Qingdao 266071, China; jinmengfei678@163.com (M.J.); lisy@qdu.edu.cn (S.L.); wuyh19@126.com (Y.W.); 18266299974@163.com (D.L.)

**Keywords:** resveratrol, nanoparticles, chitosan, alginate, delivery, UC

## Abstract

(1) Background: In the treatment of ulcerative colitis (UC), accurate delivery and release of anti-inflammatory drugs to the site of inflammation can reduce systemic side effects. (2) Methods: We took advantage of this goal to prepare resveratrol-loaded PLGA nanoparticles (RES-PCAC-NPs) by emulsification solvent volatilization. After layer-by-layer self-assembly technology, we deposited chitosan and alginate to form a three-layer polyelectrolyte film. (3) Results: It can transport nanoparticles through the gastric environment to target inflammation sites and slowly release drugs at a specific pH. The resulting RES-PCAC-NPs have an ideal average diameter (~255 nm), a narrow particle size distribution and a positively charged surface charge (~13.5 mV). The Fourier transform infrared spectroscopy showed that resveratrol was successfully encapsulated into PCAC nanoparticles, and the encapsulation efficiency reached 87.26%. In addition, fluorescence imaging showed that RES-PCAC-NPs with positive charges on the surface can effectively target and accumulate in the inflammation site while continuing to penetrate downward to promote mucosal healing. Importantly, oral RES-PCAC-NPs treatment in DSS-induced mice was superior to other results in significantly improved inflammatory markers of UC. (4) Conclusions: Our results strongly prove that RES-PCAC-NPs can target the inflamed colon for maximum efficacy, and this oral pharmaceutical formulation can represent a promising formulation in the treatment of UC.

## 1. Introduction

One goal of medicine is to find new drugs to replace drugs with limited efficacy, but drug development is a time-consuming and expensive process. The most effective sources of medicines are natural products, which can be identified through analytical techniques, such as high-performance liquid chromatography coupled with mass spectrometry (HPLC-MS) [1]. Inflammatory bowel disease (IBD) is an idiopathic disease that can cause long-term or occasional irreversible damage to the structure and function of the gastrointestinal tract [2]. At present, the analytical methods commonly used for IBD treatment include the fabric phase sorptive extraction-high performance liquid chromatography-photodiode array detection method (FPSE-HPLC-PDA), which can be used to determine residual IBD drugs [3]. It provides a guarantee for the effective treatment of IBD.

IBD includes UC and Crohn’s disease [4,5]. UC is characterized by abdominal pain, diarrhea and bleeding, which will have a negative effect on all life activities of the patient [6]. Currently, 5-aminosalicylic acids, corticosteroids and immunosuppressive drugs are used to treat colitis [7,8], but they all have different degrees of side effects. For example, mesalazine (Mesa), the first-line drug for the treatment of UC [9], may cause diarrhea, allergies, nephrotoxicity and acute pancreatiti [10,11]. Therefore, the search for targeted alternative drugs with more natural ingredients and fewer side effects has attracted much attention.

In recent years, due to the beneficial effects of polyphenols on health [12], the use of polyphenols in the field of functional foods and nutraceuticals has attracted more and more attention [13]. The plant antitoxin resveratrol (3, 5, 4′-trihydroxystilbene, RES) is one of the most studied polyphenols [14,15]. It is a nutrient and health compound extracted from natural plants such as grapes, cranberries and peanuts [16,17]. Studies have shown that RES has pharmacological effects, such as scavenging free radicals [18] and anti-inflammatory [19] and antioxidant [20] properties, which can reduce the development of IBD and improve the quality of life of patients with UC. Research by Fang Li et al. [21] also showed that RES can reduce the inflammatory symptoms and tissue damage in the dextran sodium sulfate (DSS)-induced mice colitis model. This suggests that it can be a good anti-colitis drug component. However, it is reported that RES has two configurations: cis and trans [22]. The trans-isoform has higher biological activity than the cis-isoform [12], but under sunlight and ultraviolet radiation, the trans-isoform will change to the cis-isoform. Furthermore, it has poor solubility and stability under gastrointestinal conditions [23], thus reducing its oral bioavailability and limiting its clinical application in UC.

Several drug carriers and drug delivery systems have been studied, and nano-drug delivery systems are attractive carriers. At present, researchers have encapsulated siRNA in a nano-drug delivery system to locate the intestinal inflammation site to inhibit gene expression in diseased intestinal tissue [24]. Bo Xiao loaded the anti-inflammatory tripeptide into nanoparticles (NPs) to protect it from being destroyed by the gastrointestinal tract to achieve maximum efficacy [25]. At the same time, oral administration can improve the quality of life of patients and avoid problems caused by intravenous administration, such as thrombosis and infection. Therefore, we chose an oral nano-drug delivery system to encapsulate RES. According to investigations, poly (D,L-lactide-co-glycolide) (PLGA) is a widely used copolymer approved by the FDA [26], which can be used in various medical and pharmaceutical applications [27,28]. Since the obvious initial burst release of PLGA nanoparticles (PLGA-NPs) is not conducive to drug delivery [29], we deposited chitosan and alginate on the surface of PLGA-NPs to form a polymer film (PCAC-NPs). Chitosan (CS) is a biocompatible and biodegradable polymer [30]. It has mucosal adhesion and can extend its retention time on target substrates [31]. Alginate (ALG) is a polyanionic copolymer of β-D-mannuronic acid and α-L-glucuronic acid [32], which has excellent biocompatibility, low toxicity and biodegradability [33]. Biomaterials made of ALG and CS can be degraded in the colon without being damaged by the gastric environment [34], so that NPs loaded with RES can be transported to the inflammation site of the colon, enhancing epithelial permeability and retention effect (eEPR) [25]. In addition, we characterized its physical and chemical properties (for example, particle size, Zeta potential, transmission electron microscopy, Fourier transform infrared spectroscopy and encapsulation efficiency) and further studied their ability to alleviate inflammation in the DSS-induced UC mice model.

## 2. Materials and Methods

### 2.1. Materials

PLGA (lactide: glycolide = 50:50, ester terminated, Mw = 38,000−54,000), Polyvinylalcohol (PVA, 86.5–89% hydrolyzed, viscosity 4.6–5.4 mPa·s) and resveratrol were purchased from Aladdin (Shanghai, China). Chitosan (molecular weight, 50,000–190,000; viscosity, 20–30 cP; and deacetylation 75%) and sodium alginate (low viscosity, 80,000–120,000; molecular weight, viscosity 2000 cP) were obtained from Sigma-Aldrich (St. Louis, MO, USA). Dextran sodium sulfate (DSS) was supplied by MP Biomedicals (Irvine, CA, USA). HE kits were purchased from Solarbio (Beijing, China). Male C57BL/6 mice (eight-week-old, 18–22 g) were provided by Pengyue Laboratory Animal Technology Co., Ltd. (Jinan, China), and preserved in an aseptic environment. Rhodamine B isothiocyante (RBITC) was purchased from Macklin (Shanghai, China). All other materials are analytical grade. Distilled water is used throughout the process.

### 2.2. Synthesis of RBITC Labeled CS

CS was dissolved in 100 mL acetic acid solution (2 mL acetic acid was dissolved in 100 mL water), and the pH was adjusted to 7.5 with 1 M NaOH solution. RBITC (1 mg) was dissolved in 1 mL dimethyl sulfoxide (DMSO) and added to the above solution. The reaction was stirred overnight in the dark, centrifuged and washed with water until the supernatant had no fluorescence.

### 2.3. Preparation of NPs

A total of 20 mg of PLGA and RES was mixed and dissolved in 1 mL of acetone, and then added to 4 mL of 1% (*w/v*) PVA solution and sonicated for 1 min to form an O/W emulsion. The emulsion was poured into 100 mL of distilled water and stirred for 3 h to volatilize the organic solvent to form PLGA nanoparticles (PLGA-NPs). A total of 20 mL of CS solution (1% *w/v* acetic acid dissolved in water) was dropped at a rate of 2 drops/s, stirred for 1 h and washed with water to remove unencapsulated RES and residual CS (PC-NPs). A measurement of 20 mL of ALG solution was added under gentle stirring, ultrasonic for 10 min, centrifuged at 12,000 rpm for 15 min and washed with water to remove residual ALG (PCA-NPs). Finally, 20 mL of CS solution was dropped at a speed of 2 d/s, stirred for 1 h, centrifuged at 12,000 rpm for 15 min and washed with water to remove residual CS (PCAC-NPs), and NPs were collected and freeze-dried for later use.

### 2.4. Characterization of NPs

#### 2.4.1. Particle Size and Zeta Potential

Using Zetasizer Nano ZS (Malvern Instrument, Malvern, UK), particle size was determined by dynamic light scattering (DLS) and Zeta potential of NPs determined by electrophoretic light scattering (ELS). Dimensions were measured in triplicate at room temperature, and the results were expressed as mean ± standard deviation (SD).

#### 2.4.2. TEM

Transmission electron microscope (TEM, JEM-1200 EX; JEOL, Tokyo, Japan) was used to observe the PLGA-NPs and PCAC-NPs at an acceleration voltage of 100 kV. The nanoparticle suspension was diluted and placed on a copper net to prepare the sample. After drying, the sample was observed by TEM.

#### 2.4.3. FT-IR

Fourier transform infrared spectroscopy (FT-IR, Alpha type, Bruker, Billerica, MA, USA) was used to measure the sample in the transmittance range of 4000–400 cm^−1^. The sample was fully ground into powder and then mixed with KBr to fully grind. The mixture was placed in a compression mold and pressed into a transparent sheet for determination.

#### 2.4.4. Encapsulation Efficiency (EE) and Drug Loading Capacity (LC)

RES content in the PCAC-NPs was determined indirectly by measuring the free drug content in the supernatant after the NPs were washed three times. The supernatants were combined and diluted to a 250 mL volumetric flask with water. The absorbance of the solution was measured at 306 nm by ultraviolet spectrophotometry, and the free drug content was calculated according to the standard curve. The EE and LC formulas are as follows:
$$\mathrm{EE}\%\;=\;\frac{\mathrm{Total}\;\mathrm{RES}\;\mathrm{weight}-\mathrm{Free}\;\mathrm{RES}\;\mathrm{weight}}{\mathrm{Total}\;\mathrm{RES}\;\mathrm{weight}}\times 100\%\;$$
$$\mathrm{LC}\%\;=\;\frac{\mathrm{Total}\;\mathrm{RES}\;\mathrm{weight}-\mathrm{Free}\;\mathrm{RES}\;\mathrm{weight}}{\mathrm{Total}\;\mathrm{NPs}\;\mathrm{weight}}\times 100\%\;$$


#### 2.4.5. In Vitro Drug Release

For the in vitro release studies, approximately 5 mg of RES-PCAC-NPs was suspended in 5 mL of buffered solution with different pH values (1.2, 6.8 and 7.4), filled into a dialysis bag (molecular weight 7 K) and placed in a 200 mL beaker. Then, it was incubated in a shaker at 70 rpm at 37 °C. At predetermined time intervals, 2 mL of buffer was taken to measure the absorbance, and at the same time, an equal amount of fresh buffer was added. 

In order to better simulate the release of the RES-PCAC-NPs in the body, the experiment was carried out by rotating basket method [35] in buffer solutions of pH 1.2, 6.8 and 7.4. These pH values are selected based on the normal changes in the gastrointestinal tract (GIT) from the stomach (pH ~ 1.5) to the colon (pH 7 to 7.8) [36,37]. Under 37 °C, 70 rpm incubation conditions, first, the preparation is placed in pH 1.2 buffer, and the release curve is monitored for 2 h. Then, the dialysis bag is transferred to the preheated pH 6.8 buffer solution, and the release curve is monitored for 3 h; then, it is transferred to the pH 7.4 buffer solution, and the release curve continues to be studied for 36 h. During the monitoring process, 2 mL of buffer solution is taken, and the same amount of fresh buffer solution is supplemented at the same time. The drug content was measured by spectrophotometry [38,39] at 306 nm. The cumulative drug release formula is as follows:
$$\mathrm{The}\;\mathrm{cumulative}\;\mathrm{drug}\;\mathrm{release}\%\;=\;\frac{\mathrm{Cumulative}\;\mathrm{release}\;\mathrm{drug}\;\mathrm{weight}}{\mathrm{NPs}\;\mathrm{weight}\;\times \;\mathrm{LC}\%}\;\times 100\%$$

#### 2.4.6. RES Protection

The light instability of RES limits its wide application [12]. Therefore, the protective effect of NPs on RES was studied and compared with free RES. NPs (1 mg) of PLGA-NPs, PC-NPs, PCA-NPs, PCAC-NPs loaded with RES and free RES (reference sample) were added to a glass flask in a 5% ethanol solution. After being irradiated with 254 nm ultraviolet rays at a distance of 10 cm for 120 min, samples were collected by centrifugation and freeze-dried; then, NPs were broken and dissolved in ethanol, and their content was determined by ultraviolet spectrophotometry.

### 2.5. Cell Cultures

RAW 264.7 macrophages were cultured at 37 °C in a humid atmosphere containing 5% CO_2_, and the cells were cultured in RPMI 1640 medium. Penicillin (100 U/mL), streptomycin (100 U/mL) and heat-inactivated fetal bovine serum (10%, Atlanta Biologics; Flowery Branch, GA, USA) were added to the medium.

### 2.6. Biocompatibility Assays

In vitro cytotoxicity test used standard WST-1 [40] analysis to evaluate the effect of RES-PCAC-NPs on the viability of RAW 264.7 macrophages. In short, cells were seeded into 96-well plates at a density of 5 × 103 cells/well and incubated overnight. Different concentrations of NPs (5, 10, 20, 40, 100, 200 µg/mL) were added to the plate and incubated for 24 h. Then, 10 μL of CCK-8 solution was added to each well and incubated for another 2 h in the incubator. CCK-8 was added to the cell-free medium as a blank group and untreated cells as a control group. A microplate reader was used to measure the absorbance at 450 nm.

### 2.7. Cellular Uptake

Experiments were performed using RBITC-labeled NPs to analyze the cellular uptake of NPs by fluorescence microscopy imaging. Raw 264.7 macrophages were seeded in a 24-well plate and incubated at 37 °C in a 5% CO_2_ incubator for attachment. Then, the drug-containing medium was added and the cells were incubated together for 0 h, 6 h, 12 h and 24 h. The medium was aspirated, and PBS was used to eliminate excess NPs. Then, the cells were fixed with 4% paraformaldehyde for 20 min, and DAPI was added for staining for 5 min. Images were observed using an Olympus fluorescence microscope equipped with Hamamatsu Digital Camera Orca-03G (Hamamatsu Photonics, Hamamatsu city, Japan). Images were acquired using RBITC channel and DAPI channel.

### 2.8. Animals 

Eight-week-old male C57BL/6J mice were purchased from the Jinan Pengyue Experimental Animal Breeding Co., Ltd. (Jinan, China) (Certificate SCXK20190003). The mice were maintained in temperature- and humidity-controlled equipment (25 ± 2 °C, the relative humidity is 50 ± 5%) in the photoperiod (12:12 h) in captivity and given unlimited access to standard diet and drinking water. The mice were allowed to adapt to these conditions for at least 7 days before being included in the experiment.

All animal procedures were approved by the Committee on Animals of the Qingdao University. All animal procedures were performed in accordance with the guidelines of the Committee on Animal of the Qingdao University.

#### 2.8.1. In Vivo Targeted Tracking

In order to track the NPs in GIT after oral administration, RBITC-labeled NPs were used as fluorescent probes. UC was induced for 3 days by using 3% (*wt/vol*) DSS instead of drinking water. Fluorescent labeled NPs were taken orally for 0 h, 6 h and 12 h, respectively, and the mice were imaged using an in vivo imaging system (IVIS). After that, the mice were killed immediately, the GIT tract of the mice was collected, and the IVIS (Perkin Elmer; Waltham, MA, USA) was used to obtain a more intuitive distribution of NPs in the body.

#### 2.8.2. Adhesion Experiment

The colitis tissue (6–8 mm) of DSS-treated mice and the colon (6–8 mm) of healthy mice were placed in a 24-well culture plate with the mucosal surface facing up. The wells were injected into serum-free RPMI 1640 medium, and then the RBITC-labeled NPs were added to each well. After a total of 6 h of incubation, the tissue was thoroughly washed 3 times with PBS and then quickly frozen with liquid nitrogen. The sections (5 μm) were fixed in 4% paraformaldehyde and further stained with DAPI. Images were acquired using an Olympus microscope equipped with a Hamamatsu digital camera ORCA-03G.

#### 2.8.3. DSS-Induced UC Model

The mice were randomly divided into 7 groups with 6 mice in each group. UC was induced by adding 3% (*wt/vol*) DSS (molecular weight 36–50 kDa) to drinking water. DSS solution is prepared fresh every other day. To evaluate the effect of RES-PCAC-NPs on induced colitis, mice were fed with RES-PCAC-NPs or control materials by gavage every day during the 7-day DSS treatment. In order to further explore the effect of the preparation on the treatment of colitis, the treatment was continued for 3 days. Mice were treated with RES-PCAC-NPs (the high dose is equivalent to containing RES 80 mg/kg/mice, and the low dose 40 mg/kg/mice), empty PCAC-NPs (equivalent to the quality of high-dose NPs), RES (80 mg/kg/mice) and mesalazine (Mesa, effective dose for mice based on human body weight) every day. The control mice only drank water. Weight was measured before gavage every day, and the dosage was adjusted according to the change in body weight throughout the day. The mice were observed daily and evaluated for changes in body weight and the development of clinical symptoms of colitis. Finally, the mice were sacrificed, and the spleen weight and colon length were measured. The distal colon was taken for histopathological analysis. The program has been approved by the animal protection agency.

#### 2.8.4. HE Staining and MPO Assay

HE staining can be used to evaluate the mucosal structure of the tissue sample, the signs of cell infiltration, inflammation, goblet cells and epithelial regeneration [41]. The colon tissue was fixed in 10% polymethylene formaldehyde and embedded in paraffin. HE kit is used in staining of tissue sections having a thickness of 5 μm, which are then imaged using light microscope. Myeloperoxidase (MPO) is present in neutrophils, and the number of neutrophils is quantitatively determined by analyzing the activity of the enzyme. The MPO activity unit is defined as 1 μmol of hydrogen peroxide decomposed in a reaction system at 37 °C per liter of serum, and the absorbance change at 460 nm is measured.

#### 2.8.5. Antioxidant Capacity Determination

The total antioxidant capacity test kit (ABTS method) is used to determine the total antioxidant capacity in the serum, and the absorbance is measured at 734 nm. The antioxidant capacity of the sample is directly expressed by the molar concentration of Trolox. The hydroxyl radical determination kit was used to determine the ability of serum to inhibit hydroxyl radicals, and the absorbance was measured at 550 nm. It is stipulated that each milliliter of serum is reacted at 37 °C for 1 min to reduce the concentration of H_2_O_2_ in the reaction system by 1 mmol/L as a unit of ability to inhibit hydroxyl radicals.

## 3. Results

### 3.1. Preparation and Characterization of RES-PCAC-NPs

The PLGA-NPs loaded with RES were prepared by O/W emulsion technology. Then, according to the principle that the amino group of CS and the carboxyl group of ALG can undergo electrostatic reaction, LBL self-assembly technology is applied to deposit the CS and ALG layer-by-layer on the surface of PLGA-NPs to form PCAC-NPs, as shown in Figure 1A. The particle size and Zeta potential of NPs are measured by dynamic light scattering (DLS) (Figure 1B,C). The particle size of PLGA-NPs is 226.0 ± 19.1 nm (a), the Zeta potential is −15.3 ± 2.1 mV (e) and the PDI is 0.099 ± 0.110. The deposition of CS in the first polyelectrolyte layer made the particle size change to 276.3 ± 4.3 nm (b), the potential change to 44.0 ± 11.7 mV (f) and the PDI change to 0.236 ± 0.060. The deposition of ALG in the second polyelectrolyte layer changes the size of the nanometer to 246.2 ± 20.0 nm (c), the potential to −22.0 ± 10.8 mV (g) and the PDI to 0.279 ± 0.030. The CS deposition of the third polyelectrolyte layer increased the particle size to 255.9 ± 12.0 nm (d), the potential to 13.5 ± 3.9 mV (h) and the PDI to 0.097 ± 0.095. The decrease of particle size in this process may be caused by the positive and negative charge attraction between CS and ALG. At the same time, the particle size distribution is narrow. The obvious change of Zeta potential indicates successful self-assembly on PLGA-NPs. Meanwhile, The PDI value shows that NPs are evenly distributed and tend to agglomerate less.

The morphological characteristics of the NPs were studied by TEM (Figure 2). Figure 2A shows RES-loaded PLGA-NPS with a spherical shape and smooth surface. Figure 2B,C shows the field of view of RES-loaded PCAC-NPs at 2 μm and 500 nm, respectively. It can be clearly seen that the NPs are uniformly distributed, have low aggregation and are surrounded by polyelectrolytes. This phenomenon further proves the formation of a polyelectrolyte multilayer film. In addition, by adjusting the proportional relationship between RES and PLGA, the optimal EE% of PCAC-NPs is 87.26% and LC% is 25.67%. At this time, PLGA:RES (*w/w*) is 1:1.

### 3.2. FT-IR Analysis

An FT-IR spectrophotometer was used to analyze the characteristic absorption bands of RES, PLGA, CS, ALG, RES-loaded PLGA-NPs, PC-NPs, PCA-NPs and PCAC-NPs, showing their interaction in Figure 3. The basic characteristic peaks of RES appear at ν_OH_ (3350 cm^−1^), ν_c=c_ (1650 cm^−1^) and γ_=C-H_ (965 cm^−1^). Additionally, there is the benzene ring skeleton vibration ν_c=c_ (1650 cm^−1^ and 1501 cm^−1^). The functional groups of PLGA include methyl group vibration at ν_C-H_ (2972 cm^−1^) and δ_C-H_ (1470 cm^−1^, 1380 cm^−1^) and carboxyl group vibration at ν_O-H_ (2980 cm^−1^), ν_C=O_ (1740 cm^−1^) and γ_OH_ (955 cm^−1^). The spectrum of RES-loaded PLGA-NPs showed a stretch of -OH in the range 3400~2900 cm^−1^, which was slightly offset from the previous observation and showed an increase in energy absorption. The skeleton vibration of the benzene ring did not shift significantly, indicating that the structure of the benzene ring did not change chemically. The infrared spectrum absorption peaks of CS are at ν_O-H_, ν_N-H_ (3434 cm^−1^), β_N-H_ (1596 cm^−1^), ν_C-H_ (2916 cm^−1^, 2864 cm^−1^), δ_C-H_ (1465 cm^−1^) and ν_C-O-C_ (1150 cm^−1^). The characteristic peaks of ALG are at ν_O-H_ (3400 cm^−1^), ν_C=O_ (1700 cm^−1^), γ_O-H_ (955 cm^−1^) and ν_C-O-C_ (1070 cm^−1^). When ALG is further added as a deposition layer, the main change in the infrared spectrum at this time is the formation of amide groups: ν_N-H_ (3450 cm^−1^), β_N-H_ (1620 cm^−1^), ν_C=O_ (1710 cm^−1^) and ν_C-N_ (1450 cm^−1^). These peaks indicate that the amine group of CS reacted with the carboxyl group of ALG to form an amide group. These changes indicate that RES-PCAC-NPs are not a physical mixture of various carrier materials and drugs, but a chemical combination of them. In short, RES has been successfully encapsulated in PCAC-NPs (ν: stretching vibration, γ: out-of-plane bending vibration, β: in-plane bending vibration, δ: bending vibration).

### 3.3. In Vitro Drug Release

In order to simulate the environment of the digestive system in the body, the ability of PCAC-NPs to release cargo in buffers of different pH was tested. Figure 4A shows the cumulative release of PCAC-NPs. At acidic pH, after monitoring equilibrium for 36 h, approximately 35% of the drug was released cumulatively. Under pH 6.8 and 7.4, the cumulative drug release was 45% and 55%, respectively, indicating that the drug release in simulated gastric fluid (SGF) was less than that in simulated intestinal fluid (SIF). Due to the electrostatic interaction between CS and ALG, its binding force in an acidic environment is enhanced [42], which can reduce pores and slow down the release of drugs. When entering SIF, the pH changes, and the electrostatic interaction between the two weakens, so that the pores become larger, and the drug release increases. This shows that PCAC-NPs not only can protect the drugs in SGF from being destroyed but also can release them in SIF to exert therapeutic effects.

The release of RES, PLGA-NPs and PCAC-NPs in the simulated digestive system was explored (Figure 4B), which showed that without any material protection, 80% of RES was released in the first 2 h, so that it could not better reach the colon. Without any modification, PLGA-NPs have released about 40% of the drug before reaching the administration site, which may increase the treatment time and affect the efficacy. PCAC-NPs can carry drugs and release them slowly. When passing through the stomach and small intestine, the drug loses about 20%, and the rest of the drugs can be transported to the inflammation site and released slowly, thus exerting the best effect of RES. Therefore, a polyelectrolyte film is necessary.

Figure 4D is a schematic diagram of the transportation process of PCAC-NPs from pH 1.2 to 7.4. Since the inflammation site secretes inflammatory factors and other negatively charged substances, they will attract each other with the positive surface charges of PCAC-NPs. In addition, due to the increased permeability of the inflamed tissue [34], it can adhere and accumulate on the inflamed tissue. In general, PCAC nanoshells provide good protection, which facilitates subsequent oral RES delivery.

### 3.4. RES Protection

RES appears in cis and trans isoforms. Most studies so far show that the trans isoform has higher biological activity and stability than its cis isoform [43]. However, trans RES is a highly photosensitive compound [44]. Therefore, it is demonstrated that encapsulating RES into NPs can greatly prevent the degradation of free RES, as shown in Figure 4C. After 120 min of exposure, only 19% of the trans isoforms were detected in the reference sample (unencapsulated RES), which was significantly lower than the trans isoform content in each formulation. In the uncoated PLGA-NPs, the retention percentage of RES is approximately 36%. The protective effect after coating CS was not significantly improved, which may be due to the preferential positioning of RES between the polymer chains of the coating [12]. In addition, for PCA-NPs and PCAC-NPs, the amount of trans-RES detected was 47% and 56%, respectively. This indicates that compared with coating CS alone, a larger polyelectrolyte content reduces the RES score exposed to ultraviolet radiation, thereby improving protection against conversion.

### 3.5. Cytotoxicity of RES-PCAC-NPs

For potential UC, biocompatibility is a key issue that should be carefully considered. In order to analyze the potential cytotoxicity, we used the CCK-8 kit to carry out a cytotoxicity test and studied the effect of PCAC-NPs on Raw 264.7 macrophages (Figure 5). We found that none of these concentrations significantly altered the viability of RAW 264.7 macrophages and showed no cytotoxic effects at all concentrations. These concentrations also cover the range of doses used for cell uptake studies. Taken together, our findings suggest that RES-PCAC-NPs exhibit excellent biocompatibility in vitro and therefore can be safely used as a drug delivery carrier.

### 3.6. Cellular Uptake of NPs

Effective cell uptake of NPs is a major requirement for their therapeutic efficacy. PCAC-NPs were prepared from CS labeled with RBITC and co-incubated with RAW 264.7 macrophages for 0 h, 6 h, 12 h and 24 h to allow uptake by specific cells. Afterwards, the cell nucleus was stained with blue fluorescence with DAPI to facilitate the observation of the cells’ uptake of the NPs. As shown in Figure 6, when the cells were treated with NPs at 0 h, no red fluorescence and blue fluorescence were seen to overlap, and no NPs entered the cells at this time. After incubating together for 6 h, red fluorescence was observed on the cell surface, and the cells began to take up the NPs. As the time increases to 12 h, the red fluorescence overlaps with the blue fluorescence. At 24 h, the overlap fluorescence continues to increase, showing a higher internalization efficiency. These data suggest that NPs will eventually be ingested by macrophages to function, and the uptake efficiency becomes higher with time. This is consistent with the original intention of our experiment, namely that NPs accumulate in large quantities after reaching the inflammation site and are absorbed by target cells with time to exert their effects.

### 3.7. Animals

#### 3.7.1. Distribution of NPs in the Body

The accumulation of NPs in the colon area is one of the most important features of effective nanomedicine for colon diseases, such as UC. In order to achieve this accumulation, we encapsulate the RES in PCAC-NPs. According to our above research, this nano-drug delivery system can respond to the pH value in the colon. At the same time, the positive charge on the surface of the RES-PCAC-NPs can help them adhere to the surface of the inflammation site. Then, the RES-PCAC-NPs can be delivered to the colon area and released. In order to study the biodistribution of RES-PCAC-NPs in the GIT, DSS-treated mice (UC model) were given oral RBITC-labeled PCAC-NPs. IVIS was used to examine the distribution of NPs in the GIT and in vivo targeting efficacy after 6 h and 12 h (Figure 7A). A strong fluorescent signal was observed in the small intestine after 6 h of oral NPs, and a higher content was observed in the colon after 12 h of administration. This suggests that the NPs are mainly transported to the colon. Compared with the mice in the normal group, the DSS-treated colitis mice showed significantly stronger colonic fluorescence intensity at 6 h and 12 h after the administration. In order to observe the distribution of NPs more clearly, we sacrificed the mice and took out the entire digestive system for observation (Figure 7B). Fluorescence imaging of colitis tissue showed stronger fluorescence accumulation, and it is also confirmed that RES-PCAC-NPs can smoothly reach the colonic inflammation site and accumulate infiltration, which has the potential to realize the eEPR effect.

#### 3.7.2. Adhesion and Penetration in Colonic Tissue

In order to attempt to study the adhesion and penetration of RES-PCAC-NPs in healthy colon tissue or colitis tissue, we used RBITC-labeled PCAC-NPs to co-incubate with colon tissue. Once the NPs reach colitis tissue, they must pass through the mucus layer and be internalized by target cells. As shown in Figure 8, after 6 h, only a small amount of fluorescence remained in the mucosal layer in healthy colon tissue. In contrast, many colonic epithelial cells in colitis tissues have internalized NPs and display red fluorescence. These results clearly indicate that RES-PCAC-NPs have the ability to penetrate deeply into the mucosa through the eEPR effect and are taken up by target cells (such as macrophages and colonic epithelial cells).

#### 3.7.3. Body Weight and Macroscopic Examination

Weight change is recorded as a macroscopic clinical indicator of colitis symptoms [45]. During this process, the daily weight change percentage is shown in Figure 9A. Next, we studied the therapeutic potential of RES-PCAC-NPs in vivo. C57BL/6 mice were treated with 3% DSS to induce colon inflammation, and medications were given for 7 days. After using DSS, all experimental groups showed a small increase in body weight and then a steady decrease. When the DSS treatment was stopped, we continued to intervene with different drugs for 3 days. The DSS group lost about 21% of body weight during the whole process. Mice that received Mesa, RES, empty-NPs (Empty) and low-dose (Low) and high-dose (High) RES-PCAC-NPs were observed to lose their average body weight during the entire process. Mice in the high-dose group had the lowest weight loss. In addition, after stopping the DSS treatment, the recovery effect of different preparations can be seen. After the 7th day, there was a slight recovery of body weight. It was found that oral RES-PCAC-NPs could effectively alleviate weight loss, and the high-dose group had the best effect.

The disease activity index (DAI) can assess the clinical progression of colitis after the introduction of DSS (Figure 9B) [46]. We observed that the DAI of the DSS-treated mice did not continue to increase after the DSS administration was stopped. During the whole process, the group of mice receiving high-dose RES-PCAC-NPs had the lowest DAI compared with other treatment groups. As a control, the biomaterial coated with empty-NPs also had a slight anti-inflammatory effect on DSS-induced colitis, which may be due to the biological activity of CS [47].

Since the length of the colon can directly reflect the inflammatory state of the colon [48], the mice were sacrificed, and this parameter was measured. Figure 9C,D shows that the length of the colon of the DSS group is significantly shorter than that of the healthy control group. Although the colon length of mice in all treatment groups was shorter than that of mice in the healthy control group, the colon length of mice in the RES-PCAC-NPs treatment group was much longer than that in the other preparation treatment groups. Obviously, the treatment group RES-PCAC-NPs significantly hindered the shortening of colon length. In the context of other parameters, compared with the healthy control group, the spleen weight of the mice in the DSS treatment group was significantly increased, but there was no significant difference between the other treatment groups and the healthy control group (Figure 9E). In summary, in vivo and in vitro data clearly show that the therapeutic effect of RES-PCAC-NPs is better than RES alone, which can effectively promote the recovery of UC.

#### 3.7.4. Histological and MPO Analysis

HE staining of colon tissue sections is important when studying the therapeutic effects of various treatment methods [49]. A representative picture is shown in Figure 9G. Colon tissue from the healthy control group showed no signs of inflammation or destruction. In contrast, the DSS-treated colon tissue showed obvious signs of inflammation, including epithelial destruction, loss of crypt structure, goblet cell depletion and inflammatory cell infiltration. Interestingly, the tissues from the treatment group showed less inflammation. In fact, the colon tissue of the high-dose RES-PCAC-NPs treatment group showed almost the same tissue morphology as the healthy control group, with mild damage in the epithelial layer and almost no accumulation in the immune cell spectrum. The empty-NPs treatment group had the lowest histological score, and the RES-PCAC-NPs treatment group had a better recovery effect than mesalazine. In conclusion, our histological results clearly show that the high-dose RES-PCAC-NPs treatment can effectively promote recovery from UC, which can be attributed to its targeted release in the colitis tissue. The accumulation in the inflamed colon tissue and the high absorption efficiency of macrophages also maximize the efficacy of the drug. At the same time, the therapeutic dose is also related to the efficacy.

MPO is an endogenous enzyme of mammalian granulocytes and plays a vital role in the formation of UC [41]. Here (Figure 9F), MPO activity was measured to indicate the degree of neutrophil infiltration. The MPO activity of the DSS treatment group was significantly higher than that of the healthy control group. Compared with each treatment group, the MPO activity of the mice treated with high-dose RES-PCAC-NPs was much lower.

#### 3.7.5. Antioxidant Capacity

The main sources of reactive oxygen species (ROS) include intracellular [50] (the main sources are mitochondria, endoplasmic reticulum, peroxisomes, microsomes and NOX complexes in cell membranes) and extracellular ROS-inducing agents [51] (such as radiation, pollutants and exposure to nanomaterials). Reactive oxygen species (ROS) include superoxide anions, hydrogen peroxide and hydroxyl radicals [52]. Many diseases destroy redox homeostasis, and oxidative stress represents a surge in ROS that can cause damage to cells. Excessive production of ROS in the intestinal mucosa can enhance the inflammatory response and cause mucosal damage [53]. Low levels of endogenous ROS help maintain the normal physiological functions of the body [54]. Ahmed Abdal Dayem et al. proposed that cells exposed to low concentrations of NPs show strong antioxidant defense capabilities, able to overcome oxidative stress and restore redox balance [55]. This conclusion indicates that the appropriate low concentration of nanoparticles can play a good synergistic role with drugs. In order to explore the mechanism of action of RES-PCAC-NPs, we performed hydroxyl radical scavenging experiments on mouse eyeball blood and the determination of total antioxidant capacity. As shown in Figure 10, it can be seen that the high-dose group has the strongest inhibitory ability on hydroxyl free radicals (Figure 10A). At the same time, the total antioxidant capacity also shows the same trend (Figure 10B). This indicates that reducing inflammation indicators, improving the integrity of intestinal epithelial cells and reducing the damage caused by colitis may be the role of RES-PCAC-NPs by scavenging free radicals.

## 4. Discussion

Green nanotechnology has recently become a fascinating strategy [56]. Currently, RES-loaded nanoparticles are prepared with the castor oil derivative “propenol oil” as a non-toxic solvent [57], β-lactoglobulin as the main molecule [58] and silk fibroin as the outer shell [59], and a liquid crystal structure based on glyceryl oleate encapsulates therapeutic proteins and small molecule antioxidants [60]. They have excellent delivery carrier properties, and drugs can exert their efficacy at the site of inflammation. Furthermore, related studies have shown that the membrane vesicles are a safe, biologically compatible material, which is another potential carrier for nanoherapy [61]. However, they need further research to improve the stability during gastric passage and the delay of drug release to the distal intestine. The PCAC-NPs loaded with RES that we successfully prepared are green and non-toxic and have high biocompatibility and biodegradability. At the same time, they can deliver drugs to the distal colon to make up for the aforementioned shortcomings of nanocarriers.

We have verified that the prepared RES-PCAC-NPs have excellent morphological characteristics and physical properties through various detection methods. In addition, by simulating the changes in the pH value of the gastrointestinal tract in the body, we can see that PCAC nanoshells can protect the RES from being destroyed by the stomach environment and slowly release after reaching the colon. At the same time, we also verified that PCAC nanoshells can effectively protect about 60% of the stability of RES under 2 h of light. Furthermore, the fluorescence images of fluorescently labeled RES-PCAC-NPs in cell uptake experiments and mouse experiments show that our nanoparticles have excellent targeting and aggregation properties. In summary, our nano-drug carrier system solves the gastric damage, smoothly enters the colon, adheres to the inflammation site and is further ingested by target cells into the deep part of the colon to release the drug stably.

Furthermore, the efficacy of nanoparticles was tested in DSS-induced mice, as shown in Figure 9. Our RES-PCAC-NPs show good therapeutic performance, but this is also related to the dose of the drug. The low-dose nanoparticle treatment group is obviously not as good as the high-dose nanoparticle group. However, because RES does not have the protection of the PCAC shell, the treatment effect is even more unsatisfactory. Although the drug doses of the RES group and the high-dose nanoparticle group are the same, the final result is still not as good as the low-dose nanoparticles with a shell, which may be due to the massive destruction of RES in the gastrointestinal tract. More importantly, compared with oral mesalazine enteric-coated tablets, oral RES-PCAC-NPs showed better improvement in clinical indicators of UC. Additionally, this is probably because they have the strong ability to remove active oxygen and exert an anti-inflammatory effect to improve colitis.

## 5. Conclusions

In this study, we successfully prepared RES-PCAC-NPs with a narrow particle size distribution, smooth surface, good uniformity and low agglomeration rate. In addition, we verified that RES-PCAC-NPs are less released at a low pH and slowly released at a high pH, which can protect the drug from passing through the stomach to the colon. At this time, the initial release of RES without any material protection is high. After adding a PLGA protective shell, there is a significant improvement, and a better delayed release effect is achieved after polysaccharide modification. At the cellular level, the preparation exhibits good biocompatibility and can be taken up by macrophages over time. Meanwhile, the distribution of fluorescently labeled NPs in mice shows that they have excellent aggregation and permeability in inflammation sites. Finally, various indicators in animal experiments indicate that high-dose RES-PCAC-NPs have the best ability to improve inflammation indicators in DSS-induced UC mouse models. In conclusion, this effective and highly biocompatible preparation could be developed as a promising oral platform for the treatment of UC.

## Figures and Tables

**Figure 1 nanomaterials-11-01884-f001:**
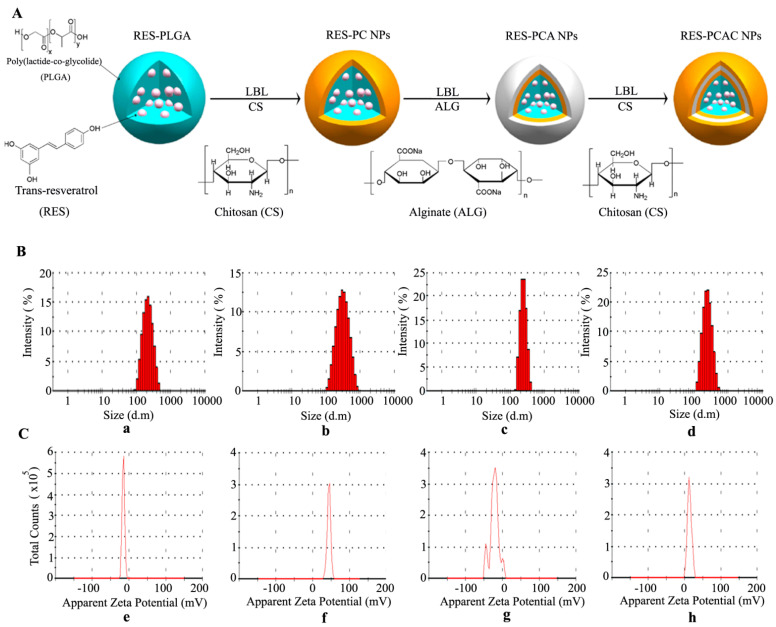
Preparation and characterization of RES-PCAC-NPs. Preparation of RES-PCAC-NPs by LBL process (**A**). The average particle size (**B**). Zeta potential (**C**) of PLGA-NPs (**a**,**e**), PC-NPs (**b**,**f**), PCA-NPs (**c**,**g**), PCAC-NPs (**d**,**h**) detected by Zetasizer Nano ZS.

**Figure 2 nanomaterials-11-01884-f002:**
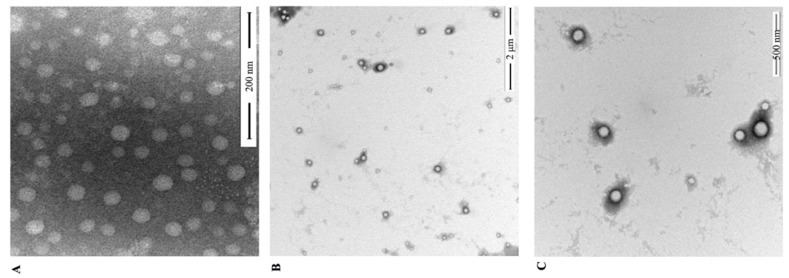
TEM images of RES-loaded PLGA-NPs (**A**); PCAC-NPs under 2μm field of view (**B**) and 500 nm field of view (**C**).

**Figure 3 nanomaterials-11-01884-f003:**
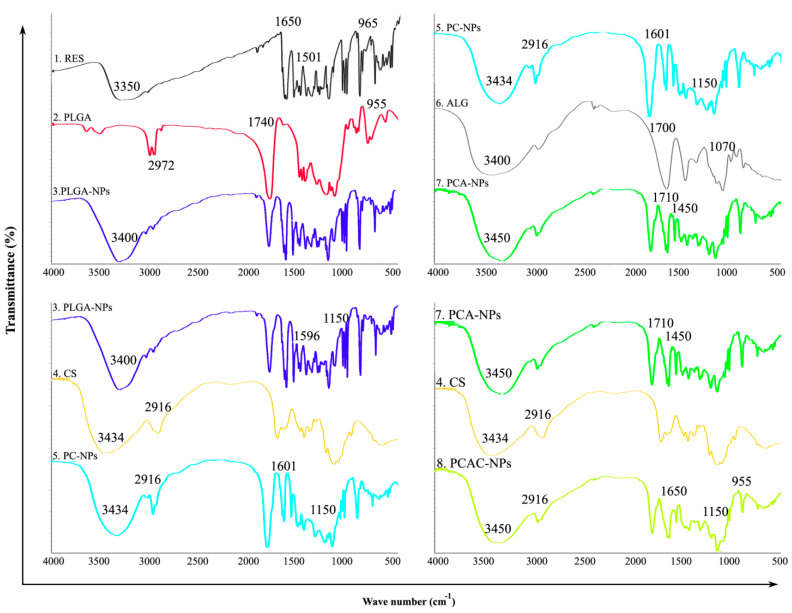
FT-IR of (1) RES, (2) PLGA, (3) RES-loaded PLGA-NPs, (4) CS, (5) RES-loaded PC-NPs, (6) ALG, (7) RES-loaded PCA-NPs and (8) RES-PCAC-NPs.

**Figure 4 nanomaterials-11-01884-f004:**
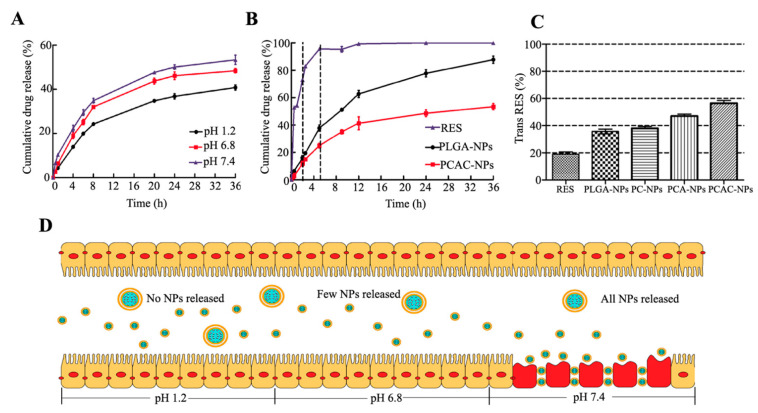
In vitro drug release: (**A**) Release curves of PCAC-NPs at pH 1.2, pH 6.8 and pH 7.4, respectively. (**B**) Release curves of RES, PLGA-NPs and PCAC-NPs in simulated digestive system. (**C**) Retention percentage of RES, PLGA-NPs, PC-NPs, PCA-NPs and PCAC-NPs exposed to ultraviolet light for 120 min. (**D**) Schematic simulation of PCAC-NPS from entering the gastric environment to being adsorbed to the inflammatory site of the colon. (In the simulated gastric environment at pH 1.2, almost no NPs release drugs. Then, the NPs enter the simulated small intestine environment at pH 6.8, and a few NPs begin to release drugs. Finally, the NPs enter the simulated colon at pH 7.4, and most of the NPs adhere to the site of inflammation and all NPs release drugs.)

**Figure 5 nanomaterials-11-01884-f005:**
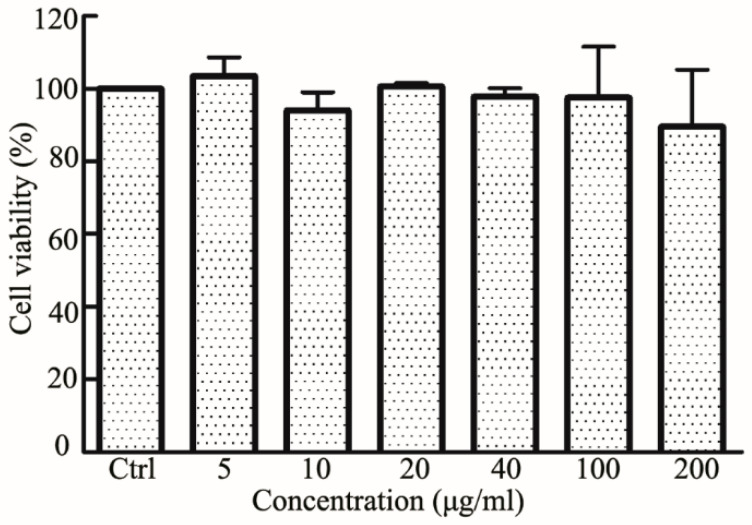
Cytotoxicity assessment of RES-PCAC-NPs at different concentrations on RAW 264.7 macrophages.

**Figure 6 nanomaterials-11-01884-f006:**
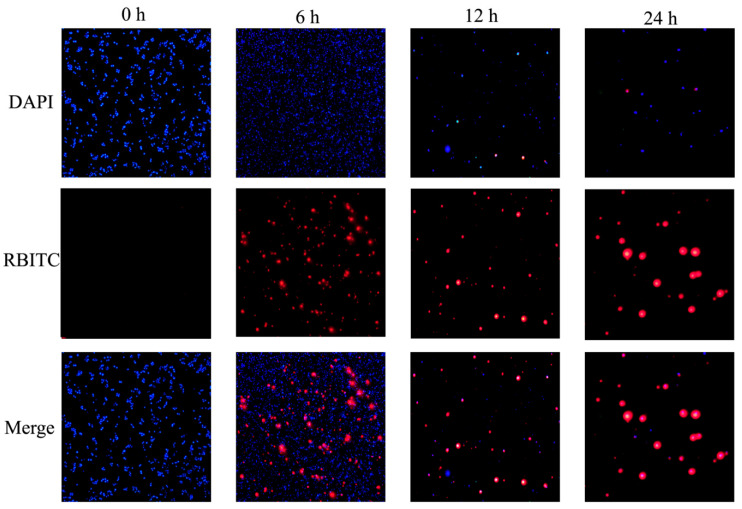
Cellular uptake of PCAC-NPs by RAW 264.7 macrophages. RAW 264.7 macrophages were treated with RBITC-labeled NPs (red) for 0, 6, 12 and 24 h, and the fixed cells were stained with DAPI to observe the nucleus (blue).

**Figure 7 nanomaterials-11-01884-f007:**
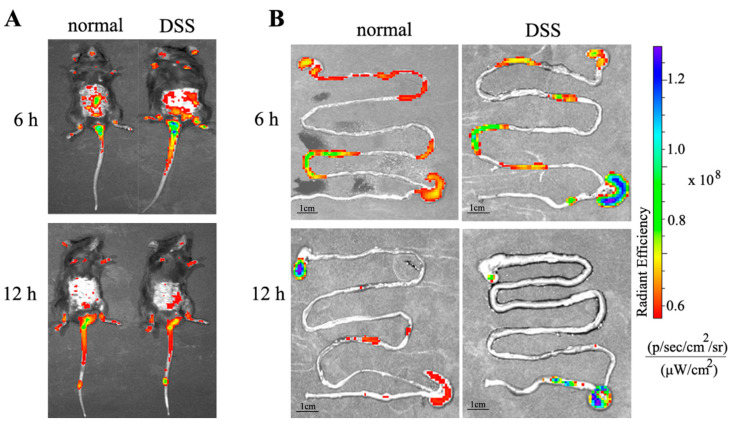
IVIS observes the fluorescence distribution in mice at 6 h and 12 h after oral administration of RES-PCAC-NPs (**A**). Near-infrared fluorescence images show the accumulation of NPs in the colon at different time points (6 h and 12 h) (**B**).

**Figure 8 nanomaterials-11-01884-f008:**
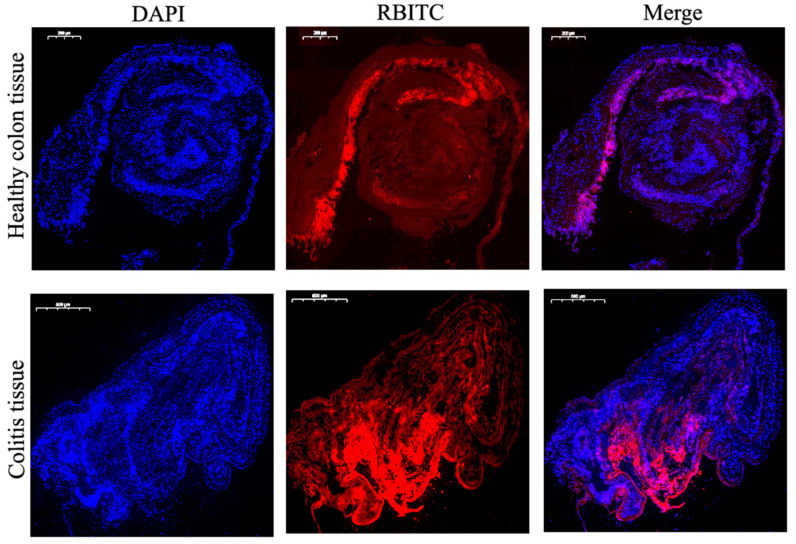
The uptake of NPs by colon tissues after 6 h of co-incubation. Healthy colon tissue and colitis tissue treated with RES-PCAC-NPs (red) and sliced, and fluorescent staining performed. The fixed tissue was stained with DAPI to observe the cell nucleus (blue).

**Figure 9 nanomaterials-11-01884-f009:**
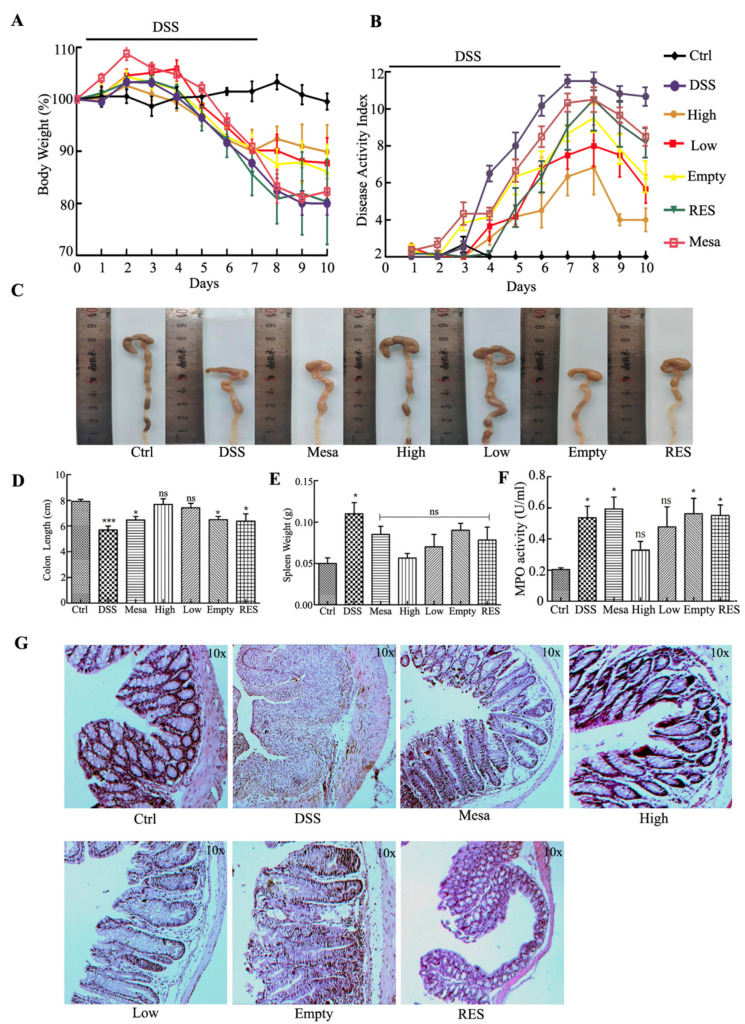
(**A**) Mice body weight over time, (**B**) disease activity index (DAI), (**C**) colon morphology from different groups of mice, (**D**) colon length, (**E**) spleen weight, (**F**) MPO activity and (**G**) DSS-treated mice fed with different preparations every day. HE staining was performed on representative colon sections after 10 days. The mice were divided into Healthy control mice group, DSS-treated mice group, Mesalazine-treated mice group, High-dose RES-PCAC-NPs-treated mice group, Low-dose RES-PCAC-NPs-treated mice group, Empty-NPs-treated mice group and RES-treated mice group. The body weight of the mice was normalized as a percentage of day 0 body weight. Each point represents the mean ± S.E.M. (n = 6). ANOVA test and Bonferroni post-test (* *p* < 0.05; *** *p* < 0.001; ns, no significance) were used to evaluate statistical significance.

**Figure 10 nanomaterials-11-01884-f010:**
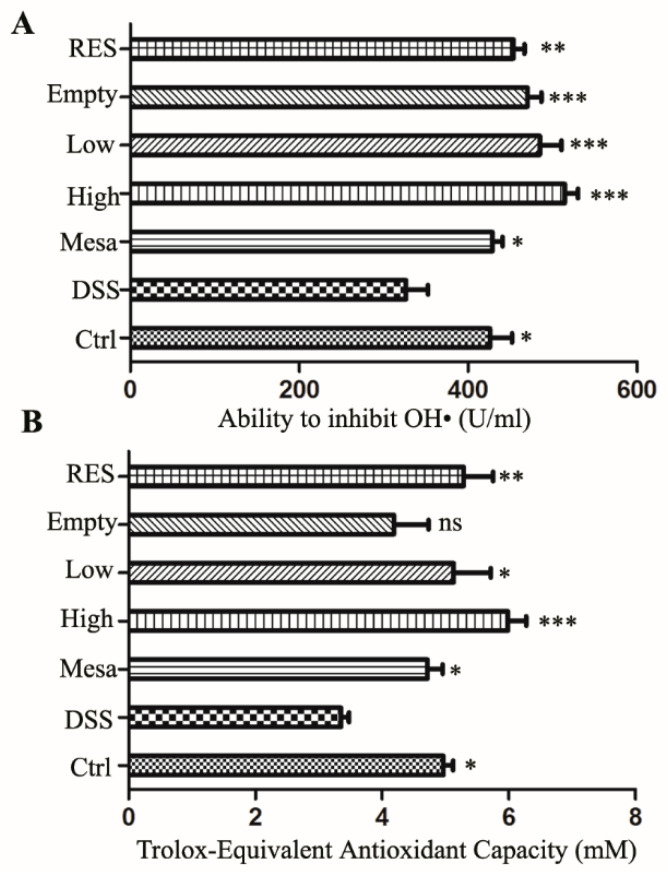
Determination of the ability to inhibit OH• (**A**) and total antioxidant capacity were determined by ABTS (**B**). Analysis of variance (ANOVA) and Bonferroni post-test (* *p* < 0.05; ** *p* < 0.01; *** *p* < 0.001; ns, no significance) were used to assess statistical significance.

## Data Availability

Data is contained within the article.

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
