# Peer review of "Construction of Chitosan/Alginate Nano-Drug Delivery System for Improving Dextran Sodium Sulfate-Induced Colitis in Mice"

_nanomaterials, 2021, doi:10.3390/nano11081884_

Round 1

Reviewer 1 Report

The authors successfully prepared chitosan/alginate nanoparticles as a drug delivery system for the treatment of ulcerative colitis. The obtained nanoparticles have a narrow size distribution, smooth surface, good uniformity, and low agglomeration rate. Tests with cells and mousses were done for evaluation of the drug release kinetics and the therapeutic effect of the proposed chitosan/alginate system.

The paper is clearly written, and I found a few minor issues for improvement:

  1. In Fig. 4d the figure caption does not give sufficient information on what is present on the figure. I could suggest a more descriptive text to be added. Why pH = 1.2 is especially emphasized?
  2. Section 3.7.5 on reactive oxygen species (ROS) and the discussion could be extended to account for recent works on this topic by other authors: (1) The Role of Reactive Oxygen Species (ROS) in the Biological Activities of Metallic Nanoparticles. Int. J. Mol. Sci. 2017, 18, 120. https://doi.org/10.3390/ijms18010120; (2) A. Cubic Liquid Crystalline Nanostructures Involving Catalase and Curcumin: BioSAXS Study and Catalase Peroxidatic Function after Cubosomal Nanoparticle Treatment of Differentiated SH-SY5Y Cells. Molecules 2019, 24, 3058. https://doi.org/10.3390/molecules24173058;

  1. The conclusion section is too short. It needs expansion to include the conclusions from the results on animals and in vivo drug release.

Author Response

Response to Reviewer 1 Comments

Comments and Suggestions for Authors:

The authors successfully prepared chitosan/alginate nanoparticles as a drug delivery system for the treatment of ulcerative colitis. The obtained nanoparticles have a narrow size distribution, smooth surface, good uniformity, and low agglomeration rate. Tests with cells and mousses were done for evaluation of the drug release kinetics and the therapeutic effect of the proposed chitosan/alginate system.

The paper is clearly written, and I found a few minor issues for improvement:

Response: Thank you very much for your kind remind, those comments are all valuable and very helpful for revising and improving our paper, as well as the important guiding significance to our researches. We have rewrited the section of 3.7.5 and Conclusion. Meanwhile, the article has careful modified in the revised manuscript.

Point 1: In Fig. 4d the figure caption does not give sufficient information on what is present on the figure. I could suggest a more descriptive text to be added. Why pH = 1.2 is especially emphasized?

Response 1: Thanks for your kind remind, we are very sorry for not clearly explaining what is on Figure 4d and we have supplement a more detailed explanation to the figure caption in Fig. 4d. Regarding why pH=1.2 is particularly emphasized, we did not want to emphasize it, but just selected a more representative pH value of the gastric environment to illustrate.

The supplementary content is as follows:

(D) Schematic simulation of PCAC-NPS from entering the gastric environment to adsorbed to the inflammatory site of the colon. (In the simulated gastric environment at pH 1.2: almost no NPs release drugs. Then the NPs enter the simulated small intestine environment at pH 6.8, and few NPs begin to release drugs. Finally, the NPs enter the simulated colon at pH 7.4 , most of the NPs adhere to the site of inflammation and all NPs release drugs.)

Point 2: Section 3.7.5 on reactive oxygen species (ROS) and the discussion could be extended to account for recent works on this topic by other authors: (1) The Role of Reactive Oxygen Species (ROS) in the Biological Activities of Metallic Nanoparticles. Int. J. Mol. Sci. 2017, 18, 120. https://doi.org/10.339,0/ijms18010120; (2) A. Cubic Liquid Crystalline Nanostructures Involving Catalase and Curcumin: BioSAXS Study and Catalase Peroxidatic Function after Cubosomal Nanoparticle Treatment of Differentiated SH-SY5Y Cells. Molecules 2019, 24, 3058. https://doi.org/10.3390/molecules24173058;

Response 2: We revisited “Section 3.7.5 on reactive oxygen species (ROS) and the discussion” and reference the publication written by Ahmed Abdal Dayem et al. and Miora Rakotoarisoa et al. We have added in "Section 3.7.5 and discussion ".

The re-modified and added content is as follows:

Section 3.7.5: The main sources of reactive oxygen species (ROS) include intracellular [50] (the main sources are mitochondria, endoplasmic reticulum, peroxisomes, microsomes and NOX complexes in cell membranes) and extracellular ROS-inducing agents [51] (such as radiation, pollutants, and exposure to nanomaterials). Reactive oxygen species (ROS) include superoxide anions, hydrogen peroxide and hydroxyl radicals [52]. Many diseases destroy redox homeostasis, and oxidative stress represents a surge in ROS that can cause damage to cells. Excessive production of ROS in the intestinal mucosa can enhance the inflammatory response and cause mucosal damage [53]. Low levels of endogenous ROS help maintain the normal physiological functions of the body [54]. Ahmed Abdal Dayem et al. proposed that cells exposed to low concentrations of NPs show strong antioxidant defense capabilities, able to overcome oxidative stress and restore redox balance [55]. This conclusion indicates that the appropriate low concentration of nanoparticles can play a good synergistic role with drugs.

Discussion: and liquid crystal structure based on glyceryl oleate encapsulates therapeutic proteins and small molecule antioxidants [60].

Point 3: The conclusion section is too short. It needs expansion to include the conclusions from the results on animals and in vivo drug release.

Response 3: Thank you for your valuable advice, we rewrote the conclusion and added about animal and in vivo drug release results. The revised content is as follows:

In this study, we successfully prepared RES-PCAC-NPs with narrow particle size distribution, smooth surface, good uniformity, and low agglomeration rate. In addition, we verified that RES-PCAC-NPs are less released at low pH and slowly released at high pH, which can protect the drug from passing through the stomach to the colon. At this time, the initial release of RES without any material protection is high. After adding PLGA protective shell, there is a significant improvement, and a better delayed release effect is achieved after polysaccharide modification. At the cellular level, the preparation exhibits good biocompatibility and can be taken up by macrophages over time. Meanwhile, The distribution of fluorescently labeled NPs in mice shows that they have excellent aggregation and permeability in inflammation sites. Finally, various indicators in animal experiments indicate that high-dose RES-PCAC-NPs have the best ability to improve inflammation indicators in DSS-induced UC mouse models. In conclusion, this effective and highly biocompatible preparation could be developed as a promising oral platform for the treatment of UC.

  1. Trachootham, D.; Alexandre, J.; Huang, P. Targeting cancer cells by ROS-mediated mechanisms: a radical therapeutic approach? Nature reviews. Drug discovery 2009, 8, 579-591, doi:10.1038/nrd2803.
  2. Vallyathan, V.; Shi, X. The role of oxygen free radicals in occupational and environmental lung diseases. Environmental health perspectives 1997, 105 Suppl 1, 165-177, doi:10.1289/ehp.97105s1165.
  3. Li, Y.; Bai, H.; Wang, H.; Shen, Y.; Tang, G.; Ping, Y. Reactive oxygen species (ROS)-responsive nanomedicine for RNAi-based cancer therapy. Nanoscale 2017, 10, 203-214, doi:10.1039/c7nr06689a.
  4. Zeng, Z.; He, X.; Li, C.; Lin, S.; Chen, H.; Liu, L.; Feng, X. Oral delivery of antioxidant enzymes for effective treatment of inflammatory disease. Biomaterials 2021, 271, 120753, doi:10.1016/j.biomaterials.2021.120753.
  5. Hoffmann, M.H.; Griffiths, H.R. The dual role of Reactive Oxygen Species in autoimmune and inflammatory diseases: evidence from preclinical models. Free radical biology & medicine 2018, 125, 62-71, doi:10.1016/j.freeradbiomed.2018.03.016.
  6. Abdal Dayem, A.; Hossain, M.K.; Lee, S.B.; Kim, K.; Saha, S.K.; Yang, G.M.; Choi, H.Y.; Cho, S.G. The Role of Reactive Oxygen Species (ROS) in the Biological Activities of Metallic Nanoparticles. International journal of molecular sciences 2017, 18, doi:10.3390/ijms18010120.
  7. 60.Rakotoarisoa, M.; Angelov, B.; Espinoza, S.; Khakurel, K.; Bizien, T.; Angelova, A. Cubic Liquid Crystalline Nanostructures Involving Catalase and Curcumin: BioSAXS Study and Catalase Peroxidatic Function after Cubosomal Nanoparticle Treatment of Differentiated SH-SY5Y Cells. Molecules (Basel, Switzerland) 2019, 24, doi:10.3390/molecules24173058.

Reviewer 2 Report

As stated on line 71 the purpose of the work was to demonstrate that a protective film of chitosan and alginate on resveratrol loaded in β-lactoglobulin nanoparticles would retard burst release under low pH gastric conditions and thus enhance delivery in the high pH colon. 

References 10 and 51 employ the same encapulation components for resveratrol but results from these studies are not discussed in the manuscript despite common areas of interest with respect to nanoproperties and clinical effectiveness. 

S Wan et al., Resveratrol-loaded PLGA nanoparticles: enhanced stability, solubility and bioactivity of resveratrol for non-alcoholic fatty liver disease therapy, Royal Society Open Science 5 (2018) 181457 addresses the critical issue of simulated digestive system resveratrol release. The conclusion from the study was that encapsulation of reservatrol in PLGA "could achieve a sustained and slow release process especially in acidic pH conditions". Even release at the small intestine pH of 6.8 was limited, about 30% at 36 hours, much lower than what is shown for PCAC-NP in Figure 4A. While comparison between studies can be misleading it does seem clear that PLGA can retain resveratrol at low pH conditions, possibly because of less erosion, without the need for covering polymer layers.

It should be noted that the lower molecular weight PLGA, 4000-15000, employed in reference 10 exhibited burst release of resveratrol at pH 6.8. The higher molecular weight used in the Wan article described above, 31000, did not.  

The dotted line indicating the end of the two hour treatment at pH 6.8 (line 146) in Figure 4B is incorrectly located at 5 hours on the x-axis.

It is not clear what drug release means. According to line 129 it was monitored by measuring the absorbance at 306 nm for the trans isomer. Figure 1A indicates that RES-PLGA consists of resveratrol encapsulated in a single PLGA nanoparticle which when coated with polymer layers yields a single RES-PCAC NP. But the schematic in Figure 4D shows smaller NPs being released from larger ones and there is no indication of any free resveratrol molecules. The illustration is supported by the statement on line 333 that negative charges at inflammatory sites will attract positively charged RES-PCAC NPs, with resveratrol not yet in contact with inflamed tissue. 

There is no control employing RES-PLGA in the animal experiments in section 3.7. According to lines 219-221 the control mice drank only water, RES means administration of pure resveratrol and empty indicates treatment with PCAC NPs not containing resveratrol. There is thus no way to evaluate the effect of the polymer films on resveratrol delivery, as compared to delivery from uncoated nanoparticles containing resveratrol, in Figures 7-10.

The manuscript is not suitable for publication in Nanomaterials since nanoparticle properties of the delivery system have been reported previously and the proposed clinical benefits of the protective chitosan/alginate layers haven't been conclusively demonstrated. 

Author Response

Response to Reviewer 2 Comments

Comments and Suggestions for Authors:

Point 1: As stated on line 71 the purpose of the work was to demonstrate that a protective film of chitosan and alginate on resveratrol loaded in β-lactoglobulin nanoparticles would retard burst release under low pH gastric conditions and thus enhance delivery in the high pH colon.

Response 1: We are very sorry that we do not understand the problem you are trying to express in this paragraph, but there is one point in the description that does not match our manuscript. We deposited chitosan and alginate on RES-loaded PLGA nanoparticles as a protective layer instead of β-lactoglobulin nanoparticles.

Point 2: References 10 and 51 employ the same encapulation components for resveratrol but results from these studies are not discussed in the manuscript despite common areas of interest with respect to nanoproperties and clinical effectiveness.

Response 2: Thank you for your instructive suggestions. We have added sentences summarizing the similarities of several articles. The revised paragraphs are as follows:

They have excellent delivery carrier properties and drugs can exert their efficacy at the site of inflammation. But they need further research to improve the stability during gastric passage and the delay of drug release to the distal intestine. The PCAC-NPs loaded with RES that we successfully prepared are green and non-toxic, and have high biocompatibility and biodegradability. At the same time, it can deliver drugs to the distal colon to make up for the aforementioned shortcomings of nanocarriers. (The first paragraph of the discussion section )

Point 3: S Wan et al., Resveratrol-loaded PLGA nanoparticles: enhanced stability, solubility and bioactivity of resveratrol for non-alcoholic fatty liver disease therapy, Royal Society Open Science 5 (2018) 181457 addresses the critical issue of simulated digestive system resveratrol release. The conclusion from the study was that encapsulation of reservatrol in PLGA "could achieve a sustained and slow release process especially in acidic pH conditions". Even release at the small intestine pH of 6.8 was limited, about 30% at 36 hours, much lower than what is shown for PCAC-NP in Figure 4A. While comparison between studies can be misleading it does seem clear that PLGA can retain resveratrol at low pH conditions, possibly because of less erosion, without the need for covering polymer layers.

Response 3: Thank you for your question. We have carefully read the article you mentioned above: Royal Society Open Science 5 (2018) 181457. However, you may not have noticed that this article is not just about PLGA nanoparticles loaded with resveratrol, it also uses another material: bovine serum albumin (BSA). Its preparation process is to transform into resveratrol-loaded PLGA nanoparticles, then emulsify them with BSA, and finally form the "PLGA-NPs" in the text (mentioned in lines 4-6 in section 2.3 of the text).

However, the function of BSA is introduced like this: BSA in the human body can be used as a carrier for various compounds. For the preparation of nanostructures, various technologies are useful, such as desolvation, emulsification, nano spray drying and NAB technology[1]. In this article, the emulsification method is used to combine PLGA with BSA to achieve a long-term release effect.

For a detailed introduction of the function of Albumin, please refer to: Albumin-based Nanoparticles for the Delivery of Doxorubicin in Breast Cancer (Introduction Section 2-4). At the same time, the nanoparticle prepared in this article concluded that among the three methods of coupling BSA, the least released was only 26% of the drug released at acidic pH after 3 days. This is consistent with the research results in the above mentioned article: encapsulation of reservatrol in PLGA "could achieve a sustained and slow release process especially in acidic pH conditions.

Finally, there is evidence for the burst release of PLGA in an acidic environment. The study by Eden Mariam Jacob et al.[2] showed that PLGA alone does not have a good protective effect under acidic pH. So the protection of the polymer layer is necessary.

Point 4: It should be noted that the lower molecular weight PLGA, 4000-15000, employed in reference 10 exhibited burst release of resveratrol at pH 6.8. The higher molecular weight used in the Wan article described above, 31000, did not. 

Response 4: This question is somewhat similar to the above question. Therefore, this question can still refer to the research of Eden Mariam Jacob et al.[2]. Because the PLGA they used has the same molecular weight as the PLGA in our experiment, the PLGA nanoparticles they prepared release about 30-40% of the drug at pH 6.8, which is similar to our results: after being transferred to pH6.8 and released for 3 hours, the RES-PLGA NPs released about 40%, while the drug release decreased by about 20% after the polymer layer was added. This is enough to show that the polymer layer is effective in slowing down the release of the drug.

However, Wan's article mentioned that PLGA-NPs with a molecular weight of 31K did not burst at pH 6.8, possibly because they used BSA to prevent ester bond hydrolysis in PLGA molecules. Since PLGA is formed by the random polymerization of two monomers-lactic acid and glycolic acid, the ester bond is broken, and the PLGA molecule will degrade.

In summary, the molecular weight of PLGA and the release of the drug cannot constitute an obvious proportional relationship. The release of the drug is also related to the hydrophobicity of the drug , organic solvent and other reasons[3].

Point 5: The dotted line indicating the end of the two hour treatment at pH 6.8 (line 146) in Figure 4B is incorrectly located at 5 hours on the x-axis.

Response 5: I'm sorry we incorrectly wrote “2 hours” in the manuscript, but in fact it was “3 hours” in the experiment. Considering that the drug stays longer in the small intestine than in the stomach, we chose to simulate the release in the small intestine for 3h. Thank you for your reminding, We have corrected it.

Point 6: It is not clear what drug release means. According to line 129 it was monitored by measuring the absorbance at 306 nm for the trans isomer. Figure 1A indicates that RES-PLGA consists of resveratrol encapsulated in a single PLGA nanoparticle which when coated with polymer layers yields a single RES-PCAC NP. But the schematic in Figure 4D shows smaller NPs being released from larger ones and there is no indication of any free resveratrol molecules. The illustration is supported by the statement on line 333 that negative charges at inflammatory sites will attract positively charged RES-PCAC NPs, with resveratrol not yet in contact with inflamed tissue.

Response 6: Regarding drug release, we measured the absorbance of trans-resveratrol and calculated the amount of drug released. This means that we can grasp the relationship between drug release and time during the entire gastrointestinal tract, and infer how much drug will not be released when it reaches the inflammation site to exert its effect. Of course, before reaching the site of inflammation, the less drug loss, the more advantageous.

Figure 1A you mentioned is the process of our preparation of the whole nanoparticle, that is, small molecule drugs (resveratrol) are encapsulated in PLGA to form the initial PLGA-NPS. Then, the chitosan/alginate is wrapped by the principle of self-assembly layer by layer to form the final RES-PCAC-NPs.

However, in Figure 4D, we are not trying to illustrate the release of smaller nanoparticles from larger nanoparticles, but we have deliberately enlarged a few in order to be able to see more clearly. Actually, what they said is a kind of nanoparticle. Figure 4D is indeed a bit confusing, so we added a more detailed explanation of the figure caption based on the reviewers’ comments.

We are really sorry that Figure 4D caused you a misunderstanding. In the figure, what we want to express is a delivery process of RES-PCAC-NPs in a simulated gastrointestinal environment. That is to say, it is well preserved at pH 1.2 and 6.8, and almost no drug is released from the nanoparticles. When it reaches the colon site at pH 7.4, the inflammation site will adsorb the positively charged nanoparticles, which facilitates the uptake of cells and releases the drug to make it work. The longer the nanoparticle stays, the more it can create conditions for cells to engulf and release drugs, which is why we want to add a polymer layer to make it stay in the inflammation site and exert a slow-release effect.

Point 7: There is no control employing RES-PLGA in the animal experiments in section 3.7. According to lines 219-221 the control mice drank only water, RES means administration of pure resveratrol and empty indicates treatment with PCAC NPs not containing resveratrol. There is thus no way to evaluate the effect of the polymer films on resveratrol delivery, as compared to delivery from uncoated nanoparticles containing resveratrol, in Figures 7-10.

Response 7: Thank you very much for your reminder. We really did not consider the completeness of the experiment and should indeed add a set of experiments. However, due to different batches of mice, different feeding times, and different operator techniques, it may bring errors to the experiment.

However, it is possible to guess the result of the final experiment. The efficacy of PLGA-NPs is denied based on simulated in vitro drug release experiments, and continued verification in animal experiments is not necessary.

The role of our polymer layer is to allow NPs to pass through the stomach, adhere to and accumulate at the inflammation site, and be taken up by the cells to release drugs to exert their drug effects. At this time, we observed the effects of high and low doses of drugs and the shell on colitis, and did not make a focused comparison between RES-PCAC-NPs and PLGA-NPs, because we did not want to explore the gap between the two carriers.

Point 8: The manuscript is not suitable for publication in Nanomaterials since nanoparticle properties of the delivery system have been reported previously and the proposed clinical benefits of the protective chitosan/alginate layers haven't been conclusively demonstrated.

Response 8: Based on the properties of nanoparticles in the delivery system reported, we conducted a preliminary literature review. However, unlike us, other similar nanoparticle studies are only stop at the pre-preparation and detection stage. We are the only nano-drug carrier platform prepared for the treatment of colitis, and we have carried out more comprehensive work on the nano-drug carrier system. Such as the detection of basic properties, the evaluation of the cell level and the animal level, and the detection of drug efficacy. This is very consistent with our manuscript submitted in "Nanomaterials: Applications of Nanoparticles in Biology and Medicine".

  1. Prajapati, R.; Garcia-Garrido, E.; Somoza, Á. Albumin-Based Nanoparticles for the Delivery of Doxorubicin in Breast Cancer. Cancers 2021, 13, doi:10.3390/cancers13123011.
  2. Jacob, E.M.; Borah, A.; Pillai, S.C.; Kumar, D.S. Garcinol Encapsulated Ph-Sensitive Biodegradable Nanoparticles: A Novel Therapeutic Strategy for the Treatment of Inflammatory Bowel Disease. Polymers 2021, 13, doi:10.3390/polym13060862.
  3. Oseni, B.A.; Azubuike, C.P.; Okubanjo, O.O.; Igwilo, C.I.; Panyam, J. Encapsulation of Andrographolide in poly(lactide-co-glycolide) Nanoparticles: Formulation Optimization and in vitro Efficacy Studies. Frontiers in bioengineering and biotechnology 2021, 9, 639409, doi:10.3389/fbioe.2021.639409.

Reviewer 3 Report

The paper "Construction of chitosan/alginate nano-drug delivery system for improving dextran sodium sulfate-induced colitis in mice" is a good work. In my opinion the topic fits with Journal aim and scope and the text is well organized.

I suggest to accept the paper pending revisions, particularly:

  • in introduction section add a paragraph reporting the recent analytical approaches applied in this field for IBD and related references (e.g. Journal of Chromatography B: Analytical Technologies in the Biomedical and Life SciencesVolume 1084, Pages 53 - 631 May 2018; Biomedical Chromatography (ISSN:1099-0801 online), 26(3), 283-300, 2012; and Frontiers in Microbiology sect. Systems Microbiology (ISSN: 1664-302X online), 8, article 1040, 1-10, 2017)
  • in my opinion for In vitro drug release is better to use the HPLC procedure (preferably validated) in order to avoid interferences that could bias the results
  • please clarify how is evaluated the entrapment efficiency

Author Response

Response to Reviewer 3 Comments

Comments and Suggestions for Authors:

The paper "Construction of chitosan/alginate nano-drug delivery system for improving dextran sodium sulfate-induced colitis in mice" is a good work. In my opinion the topic fits with Journal aim and scope and the text is well organized.

I suggest to accept the paper pending revisions, particularly:

Response: Thank you very much for your careful work and kind remind, those comments are all valuable and very helpful for revising and improving our paper, as well as the important guiding significance to our researches.

Point 1: In introduction section add a paragraph reporting the recent analytical approaches applied in this field for IBD and related references (e.g. Journal of Chromatography B: Analytical Technologies in the Biomedical and Life SciencesVolume 1084, Pages 53 - 631 May 2018; Biomedical Chromatography (ISSN:1099-0801 online), 26(3), 283-300, 2012; and Frontiers in Microbiology sect. Systems Microbiology (ISSN: 1664-302X online), 8, article 1040, 1-10, 2017)

Response 1: The reviewer is absolutely right that we need to give more additional references of previous publication about the recent analytical approaches applied in this field for IBD. The additions are as follows:

One goal of medicine is to find new drugs to replace drugs with limited efficacy, but drug development is a time-consuming and expensive process. The most effective sources of medicines are natural natural products, which can be identified through analytical techniques, such as high-performance liquid chromatography coupled with mass spectrometry (HPLC-MS)[1]. Inflammatory bowel disease (IBD) is an idiopathic one that can cause long-term or occasional irreversible damage to the structure and function of the gastrointestinal tract [2]. At present, the analytical methods commonly used for IBD treatment include fabric phase sorptive extraction-high performance liquid chromatography-photodiode array detection (FPSE-HPLC-PDA) method, which can be used to determine residual IBD drugs[3]. It provides a guarantee for the effective treatment of IBD.

Furthermore, related studies have shown that the membrane vesicles are a safe, biological compatible material, which is another potential carrier for nanoherapy [61].( Lines 7-8 of the first paragraph of the discussion section)

Point 2: In my opinion for In vitro drug release is better to use the HPLC procedure (preferably validated) in order to avoid interferences that could bias the results

Response 2: Thank you for your careful work, we absolutely agree that using the HPLC procedure can make the results more accurate and it is also a good choice. However, considering the photosensitivity of resveratrol, there may be deviations in the waiting process.

Therefore, we chose to use a relatively fast and convenient detection method: spectrophotometry. Although the error of this method may be larger than that of HPLC method, its measurement is relatively stable, and we have carried out multiple repeated experiments to keep it within a certain range of deviation.

 Finally, we also refer to other researchers' methods of determining drug content, indicating that this method is also feasible. For example, Nieves Iglesias[38] and Vanna Sanna[39] used spectrophotometry to determine the content of drugs. At the same time, we also added these references to the manuscript to make it more convincing.  

Point 3: Please clarify how is evaluated the entrapment efficiency

Response 3: Encapsulation efficiency (EE) refers to the percentage of the encapsulated substance (such as a drug) in the total amount of the drug in the liposome suspension. It is an important indicator for the quality control of liposomes and nanoparticles, and reflects the extent to which the drug is encapsulated by the carrier.

EE% = actual drug-loaded quality/ theoretical drug-loaded quality x100%. Generally speaking, the higher the EE%, the more the actual drug load. When the EE% is closer to 100%, it indicates that the loss of the drug is less, and it can also reflect that the preparation is good.

 In the process of preparing our nanoparticles, we explored the relationship between the drug and PLGA ratio to minimize the drug loss, and the best EE% reached 87%. Therefore, it can be explained that the nanoparticles prepared by our method have better encapsulation efficiency.

  1. Locatelli, M.; Governatori, L.; Carlucci, G.; Genovese, S.; Mollica, A.; Epifano, F. Recent application of analytical methods to phase I and phase II drugs development: a review. Biomedical chromatography : BMC 2012, 26, 283-300, doi:10.1002/bmc.1674.
  2. Xavier, R.J.; Podolsky, D.K. Unravelling the pathogenesis of inflammatory bowel disease. Nature 2007, 448, 427-434, doi:10.1038/nature06005.
  3. Kabir, A.; Furton, K.G.; Tinari, N.; Grossi, L.; Innosa, D.; Macerola, D.; Tartaglia, A.; Di Donato, V.; D'Ovidio, C.; Locatelli, M. Fabric phase sorptive extraction-high performance liquid chromatography-photo diode array detection method for simultaneous monitoring of three inflammatory bowel disease treatment drugs in whole blood, plasma and urine. Journal of chromatography. B, Analytical technologies in the biomedical and life sciences 2018, 1084, 53-63, doi:10.1016/j.jchromb.2018.03.028.

  1. Iglesias, N.; Galbis, E.; Díaz-Blanco, M.J.; Lucas, R.; Benito, E.; de-Paz, M.V. Nanostructured Chitosan-Based Biomaterials for Sustained and Colon-Specific Resveratrol Release. International journal of molecular sciences 2019, 20, doi:10.3390/ijms20020398.
  2. Sanna, V.; Siddiqui, I.A.; Sechi, M.; Mukhtar, H. Resveratrol-loaded nanoparticles based on poly(epsilon-caprolactone) and poly(D,L-lactic-co-glycolic acid)-poly(ethylene glycol) blend for prostate cancer treatment. Molecular pharmaceutics 2013, 10, 3871-3881, doi:10.1021/mp400342f.

  1. Grande, R.; Celia, C.; Mincione, G.; Stringaro, A.; Di Marzio, L.; Colone, M.; Di Marcantonio, M.C.; Savino, L.; Puca, V.; Santoliquido, R.; et al. Detection and Physicochemical Characterization of Membrane Vesicles (MVs) of Lactobacillus reuteri DSM 17938. Frontiers in microbiology 2017, 8, 1040, doi:10.3389/fmicb.2017.01040.

Round 2

Reviewer 2 Report

No comments

Reviewer 3 Report

All requested changes were reported in teh revised version. In my opinion the paper can be now accepted for publication